# Investigation of the Correlation between Enterovirus Infection and the Climate Factor Complex Including the Ping-Year Factor and El Niño-Southern Oscillation in Taiwan

**DOI:** 10.3390/v16030471

**Published:** 2024-03-20

**Authors:** Hsueh-Wen Yu, Chia-Hsuan Kuan, Liang-Wei Tseng, Hsing-Yu Chen, Meg-Yen Tsai, Yu-Sheng Chen

**Affiliations:** 1Department of Chinese Acupuncture and Traumatology, Center for Traditional Chinese Medicine, Chang Gung Memorial Hospital, No. 123, Dinghu Rd., Gueishan Dist., Taoyuan City 333423, Taiwan; b9905026@cgmh.org.tw (H.-W.Y.); gh70106@gmail.com (C.-H.K.); frank.tseng.sr@gmail.com (L.-W.T.); 2Taiwan Huangdi-Neijing Medical Practice Association (THMPA), Taoyuan City 330032, Taiwan; 3Department of Chinese Internal Medicine, Center for Traditional Chinese Medicine, Chang Gung Memorial Hospital, No. 123, Dinghu Rd., Gueishan Dist., Taoyuan City 333423, Taiwan; 8705016@cgmh.org.tw; 4Pingzhen Fengze Chinese Medicine Clinic, No. 65, Sec. 2, Yanping Rd., Pingzhen Dist., Taoyuan City 324005, Taiwan; spiritfly903@gmail.com; 5School of Traditional Chinese Medicine, Chang Gung University, No. 259, Wen-Hwa 1st Rd., Gueishan Dist., Taoyuan City 333323, Taiwan

**Keywords:** enterovirus, ping-year factor, Yunqi theory, El Niño-Southern Oscillation, ENSO, outbreak

## Abstract

Enterovirus infection and enterovirus infection with severe complications (EVSC) are critical issues in several aspects. However, there is no suitable predictive tool for these infections. A climate factor complex (CFC) containing several climate factors could provide more effective predictions. The ping-year factor (PYF) and El Niño-Southern Oscillation (ENSO) are possible CFCs. This study aimed to determine the relationship between these two CFCs and the incidence of enterovirus infection. Children aged 15 years and younger with enterovirus infection and/or EVSC were enrolled between 2007 and 2022. Each year was categorized into a ping-year or non-ping-year according to the PYF. Poisson regression was used to evaluate the associations between the PYF, ENSO, and the incidence of enterovirus infection. Compared to the ping-year group, the incidence rate of enterovirus infection, the incidence rate of EVSC, and the ratio of EVSC in the non-ping-year group were 1.24, 3.38, and 2.73 times higher, respectively (*p* < 0.001). For every one-unit increase in La Niña, the incidence rate of enterovirus infection decreased to 0.96 times (*p* < 0.001). Our study indicated that CFCs could be potential predictors for enterovirus infection, and the PYF was more suitable than ENSO. Further research is needed to improve the predictive model.

## 1. Introduction

Enterovirus belongs to the *Picornaviridae* family, characterized by a high mutation and recombination rate [1]. Currently, over one hundred serotypes have been identified, and a new classification system has been developed, grouping enteroviruses into twelve species, A to L. Human enterovirus accounts for A, B, C, and D and include poliovirus, coxsackievirus A and B (CA/CB), echovirus (ECHO), and other enterovirus [2]. Enterovirus infection is mainly spread by direct contact with stool or respiratory secretions from an infected person or on contaminated surfaces. Clinical manifestations range from asymptomatic infections and mild symptoms, such as hand, foot, and mouth disease and herpangina, to severe complications including aseptic meningitis, viral encephalitis, myocarditis, and acute hemorrhagic conjunctivitis [3]. In Taiwan, enterovirus infection with severe complications is classified into the third category of communicable diseases. The most prevalent viral types of enteroviruses are CA, CB, ECHO, EV-D68, and EV-71. Other enteroviruses may include few rhinovirus and emerging/re-emerging enterovirus types like parechovirus, which used to be classified as enterovirus but are now differentiated as an independent genus in the *Picornaviridae* family [4]. The monitoring of enterovirus infections has always been an important goal of the Centers for Disease Control (CDC) of the Ministry of Health and Welfare. The current CDC established a real-time monitoring system and related statistical data to analyze the geographical [5], temporal, and medical cost [6] characteristics of enterovirus infection occurrence. Epidemiologically, enterovirus infection often occurs in the summer and early autumn [7].

Previous studies have found a correlation between climate factors and the prevalence of enterovirus infections, with several climate factors, including temperature, humidity, rainfall, wind speed, sunshine, and air pressure, being identified as important factors. For example, temperature has been widely recognized as one of the most important climate factors that affect the prevalence of enterovirus infection [8,9,10,11]. One study found a positive correlation between the incidence of hand, foot, and mouth disease (HFMD) and a 1 °C increase in temperature [8], while other studies showed that temperature has a delayed impact on enterovirus prevalence ranging from several days to weeks [9,10,12]. However, some studies have found a nonlinear inverted V-shaped relationship between HFMD and temperature [13], while others have found no statistically significant correlation [14,15]. These studies were conducted in different Asian countries [9,10,16,17]. Considering the varying climate conditions across different regions, it is not appropriate to rely on a single climate factor. Instead, a climate factor complex (CFC) that comprises several climate factors should be used to identify the suitable range for the incidence of enterovirus infection. Under different types of CFC, the relationship between enterovirus infections and climate factors is accordingly diverse.

We assume the ping-year factor (PYF), an indicator in the Yunqi theory of Huangdi’s Internal Classic (Huang Di Nei Jing), to be a kind of CFC. Huangdi’s Internal Classic systematically discusses the relationship between climate change, the astronomy cycle, calendars, the human body, disease, and the environment. Yunqi theory is one of the theories in Huangdi’s Internal Classic that is based on ten Celestial Stems (Heavenly Stems) and twelve Terrestrial Branches (Earthly Branches) [18,19,20]. Depending on the different characteristics from the interaction of the Celestial Stems and Terrestrial Branches of each year, the climate changes and the manifestations of corresponding diseases are affected. Previous research has found a significant positive correlation between the PYF and the trend of major epidemic events documented in the Ming and Qing dynastic historical records (1371–1911) [7]. According to the definition of the PYF, every year can be categorized into either a ping-year (PY) or a non-ping-year (NPY). The term “ping-year” (PY) originates from the Chinese word “氣之平” (píng of Qi), which conveys the meanings of stability and peace of Qi. As a result, “ping-year (unharmed-year)” refers to a year with a lower occurrence of epidemics and unfavorable for the spread of diseases, whereas “non-ping-year (harmful-year)” represents a year with a higher occurrence of epidemics and favorable for the spread of diseases. Based on the above, we speculate that the PYF could be a potential method for predicting the incidence of enterovirus infection.

Another potential CFC for predicting the incidence of enterovirus infection is the El Niño-Southern Oscillation (ENSO). ENSO is a quasiperiodic phenomenon characterized by alternating warm El Niño and cold La Niña conditions, representing the strongest year-to-year fluctuation of the global climate system [21]. During the El Niño phase, the sea surface temperatures in the central and eastern tropical Pacific Ocean increase, causing air to rise and surface air pressure to decrease. This leads to increased rainfall, particularly in the usually dry eastern Pacific region (such as Peru), where heavy rains and the risk of flooding are more likely to occur. On the other hand, the changes in the rainy tropical western Pacific region (such as Indonesia and Malaysia) are opposite to those in the eastern Pacific. Not only does rainfall decrease, but there is also a possibility of drought. The La Niña phase exhibits a trend opposite to that of the El Niño phase. These changes in atmospheric and oceanic circulation can affect the global climate and human activities. ENSO may also impact the spread of certain diseases, but not all diseases are affected [22,23,24,25,26,27,28]. Evidence suggests that ENSO has a stronger effect on malaria and cholera than on other mosquito-borne and rodent-borne diseases [28]. However, a limited number of studies have investigated the effect of ENSO on enterovirus infection. Only one study indicated that there is a short-term effect of ENSO, with a positive correlation between a high Southern Oscillation Index (SOI) and enterovirus (SOI = 45, relative risk = 1.66) [22]. If prolonged periods of positive SOI values exist, the La Niña phase is considered.

Our study analyzed the correlation between the incidence of enterovirus infection and two different kinds of CFCs, i.e., the PYF and ENSO, in recent decades, aiming to predict epidemic trends and provide guidance for early warning and monitoring in the future.

## 2. Materials and Methods

### 2.1. Data Collection

Annual cases of enterovirus diagnosed in the emergency department and annual cases of enterovirus infection with severe complications (EVSC) were both collected from the Taiwan National Infectious Disease Statistics System on the website of the Taiwan Centers for Disease Control (https://nidss.cdc.gov.tw/en/Home/Index?op=1, accessed on 30 March 2023). The data of annual enterovirus cases were transferred from the National Health Insurance research database of Taiwan by using ICD-9-CM/ICD-10 codes. Herpangina and HFMD (ICD-9-CM of 074.3 or 074.0, and ICD-10 of B08.4 or B08.5, respectively) were diagnosed based on clinical symptoms and signs, with or without laboratory examination [29]. The available enterovirus statistical data ranged from 2007 to 2022. Cases of individuals over 15 years of age were excluded due to the extremely low incidence rate among these individuals [30].

All annual cases of EVSC meet both the clinical and laboratory criteria.

#### 2.1.1. Clinical Criteria: Any One of the Following Three Criteria

Typical symptoms of hand, foot, and mouth disease or herpangina, with the presence of myoclonic jerks, or complications such as encephalitis, acute flaccid paralysis syndrome, acute hepatitis, myocarditis, acute cardiomyopathy, heart and lung failure, and other severe conditions.Absence of hand, foot, and mouth disease or herpangina, but with respiratory infection symptoms and suspected enterovirus infection accompanied by brainstem encephalitis or acute flaccid myelitis.Infants under three months of age who presented with symptoms of sepsis, such as myocarditis, hepatitis, encephalitis, thrombocytopenia, and multiple organ failure, excluding other common bacterial infections.

#### 2.1.2. Laboratory Criteria: Any One of the Following Three Criteria

Clinical specimens (throat swabs or throat washes, feces or rectal swabs, cerebrospinal fluid or vesicle fluid, etc.) were isolated and identified as enterovirus.Molecular biological nucleic acid testing of clinical specimens was positive.Serological antibody testing was positive (referring to specific IgM antibodies against enterovirus type 71 in serum)

The virologic surveillance data reported from the Taiwan Centers for Disease Control includes CA, CB, ECHO, EV71, EV-D68, and other enteroviruses. “Other enteroviruses” refers to all serotypes that were not among the top 5 mentioned and were not individually listed [31]. Rhinovirus, human parechoviruses, and other emerging/re-emerging enterovirus types have been excluded once identified [32]. The methods used by the contracted laboratories of the Tw-CDC for enterovirus infections include viral culture, immunofluorescence assay, and polymerase chain reaction (EV CODEHO*p* < RT–snPCR or pan-enterovirus real-time RT-PCR, target gene: 5’UTR and VP1). If the type of enterovirus could not be differentiated, further Sanger sequencing would be arranged [33,34]. The number of people younger than 15 in a year was collected from the Department of Household Registration, M.O.I., Taiwan (https://www.ris.gov.tw/app/en/3911, accessed on 30 March 2023)

### 2.2. Incidence Rate Calculation

For evaluating enterovirus infection and EVSC separately [35], we designed three formulae as follows:1.Incidence rate of enterovirus infection
$$\frac{number\;of\;patients\;diagnosed\;with\;enterovirus\;infection\;in\;a\;year}{total\;population\;of\;children\;under\;15\;years\;of\;age\;in\;a\;year}$$

2.Incidence rate of EVSC


$$\frac{number\;of\;patients\;diagnosed\;with\;EVSC\;in\;a\;year}{total\;population\;of\;children\;under\;15\;years\;of\;age\;in\;a\;year}$$


3.Ratio of EVSC


$$\frac{number\;of\;patients\;diagnosed\;with\;EVSC\;in\;a\;year}{number\;of\;patients\;diagnosed\;with\;enterovirus\;infection\;in\;a\;year}$$


### 2.3. Grouping

#### 2.3.1. Ping-Year Factor (PYF)

In the Yunqi theory, the Celestial Stems govern the Major Yun, while the Terrestrial Branches govern the type of Major Qi each year (Table 1 and Table 2). PYF stands for a certain combination of Celestial Stems and Terrestrial Branches. According to PYF, each year can be categorized as a ping-year (PY) or non-ping-year (NPY) [7]. In a cycle of 60 years, there is a total of 28 PYs and 32 NPYs (Table 3).

#### 2.3.2. ENSO

We used the Oceanic Niño Index (ONI) to represent the activity level of ENSO. The ONI tracks the continuous 3-month average sea surface temperatures compared to the 1981–2010 average in the east-central tropical Pacific between 120° and 170° W. Data on the ONI were extracted from the Climate Prediction Center of the National Oceanic and Atmospheric Administration (CPC-NOAA) (http://www.cpc.ncep.noaa.gov/products/analysis_monitoring/ensostuff/ensoyears.shtml, accessed on 30 March 2023). ONI values of +0.5 or higher (positive ONI) were considered El Niño conditions, values between +0.5 and −0.5 were considered neutral conditions, and values of −0.5 or lower (negative ONI) were considered La Niña conditions. The threshold was further broken down into different levels as weak (0.5 to 0.9 sea surface temperature (SST) anomaly), moderate (1.0 to 1.4), strong (1.5 to 1.9), and very strong (≥2.0) events (https://ggweather.com/enso/oni.htm?fbclid=IwAR0qmJviclKJjBDKn2NyS1_7a-jOO0hhkw9LaHELTVep3Qm36azin9oBeMc, accessed on 30 March 2023) (Table 4).

### 2.4. Statistical Analysis

As the annual count of enterovirus infection commonly followed a Poisson distribution, a generalized additive model (GAM) with a log link and Poisson auto-regression was applied to investigate the effects of climate factor complex on enterovirus infection [36,37]. With Poisson data assumption, the model we applied in this study was decomposed into components including PYF and ENSO:$$Y_{t}\;\sim \;Poisson(\mu_{t})$$
$$Ln\left(\mu_{t}\right)=\alpha +\sum \beta_{1}\;P_{t}+\sum \beta_{2}\;E_{t}$$
where $Y_{t}$ is the annual count of enterovirus infection in year *t*; *α* is the intercept; *P_t_* is the ping-year factor of year *t*; *E_t_* is the ENSO level of year *t*; *β* is the regression coefficient.

Poisson regression models were used to test for differences between the two groups (PY and NPY) in the following three outcomes: incidence rate of enterovirus infection, incidence rate of EVSC, and ratio of EVSC. Due to consecutive changes in ENSO, differences between the different activities were tested by Poisson regression models. Instead of El Niño conditions, we assumed La Niña conditions to be positively related to the incidence of enterovirus infection according to a previous study [22].

## 3. Results

### 3.1. The PYF and Enterovirus Infection

According to the Yunqi theory, from 2007 to 2022, there was a total of eight PYs and eight NPYs (Table 5). Our results showed that the incidence rate of enterovirus infection was 1.24 times higher in the NPY than in the PY (95% Cl 1.23 to 1.24, *p* < 0.001). The incidence rate of EVSC was 3.38 times higher in the NPY than in the PY (95% Cl 2.88 to 3.99, *p* < 0.001). The ratio of cases of EVSC among enterovirus patients in the NPY was 2.73 times higher than that in the PY (95% Cl 2.33 to 3.23, *p* < 0.001) (Table 6).

Figure 1 and Table 5 describe the annual number of cases, total population, incidence rates of both enterovirus infections and EVSCs, and the ratio of EVSCs over the years. In terms of the incidence rate of enterovirus infection, two of the top three years with the highest incidence rates were NPYs (2013 and 2016), while two of the three years with the lowest incidence rates were PYs (2020 and 2021). Among the eight years with an incidence rate higher than the annual average (62.72 people per 10,000 population), five were NPYs; among the eight years with an incidence rate lower than the average, five were PYs.

In the PY group, except for 2010 (118.87 cases per 100,000 population), 2017 (74.35 cases per 100,000 population), and 2019 (89.71 cases per 100,000 population), the incidence rates of enterovirus infection were lower than the annual average incidence rate (62.72 people per 10,000 population). In the NPY group, except for 2008 (61.98 cases per 100,000 population), 2011 (55.16 cases per 100,000 population), and 2022 (4.24 cases per 100,000 population), the incidence rates of enterovirus infection were higher than the annual average incidence rate.

In terms of enterovirus infection with severe complications, two of the top three years were NPYs (2008 and 2012), while two of the lowest three years were PYs (2020 and 2021). There were four years with incidence rates higher than the annual average incidence rate (0.134 cases per 10,000 population), including three NPYs (2008, 2011, and 2012) and one PY (2019). There was a total of twelve years with incidence rates lower than the annual average incidence rate, including five NPYs and seven PYs. The ratio of EVSCs was similar in that the ratio of the same four years (2008, 2011, 2012, and 2019) was higher than the average.

Sensitivity analysis was also performed to test the robustness of our results. Taking into consideration that the number of cases may have had outliers in certain years, we conducted a sensitivity analysis, which revealed that the statistical results obtained were similar when each individual year was excluded (See Appendix A).

### 3.2. ENSO and Enterovirus Infection

Our results indicated that for every one-unit increase in the activity of La Niña, the incidence rate of enterovirus infection decreased to 0.96 times (95% Cl 0.96 to 0.97, *p* < 0.001), which means that the higher the La Niña level in ENSO, the lower the incidence rate of enterovirus infection (Table 6). There was an opposite trend in both the incidence rate of EVSCs and the ratio of EVSCs. The ratio of EVSCs increased by a factor of 1.06 (95% Cl 1.06 to 1.10, *p* = 0.008) for every one-unit increase in the activity of La Niña, while the *p*-value for the incidence rate of EVSCs was not significant.

### 3.3. The PYF, ENSO, and Enterovirus Infection

By combining the PYF and ENSO, we could eliminate their influence on each other and thus standardize the factors. Regarding the incidence rates of enterovirus infections and EVSCs and the ratio of EVSCs, the PYF showed a positive relationship when analyzed both separately and together. The incidence rate of enterovirus infection was 1.29 times higher in the NPY than in the PY (95% Cl 1.28 to 1.29, *p* < 0.001). The incidence rate of EVSC was 2.38 times higher in the NPY than in the PY (95% Cl 2.00 to 2.83, *p* < 0.001). The ratio of EVSC in the NPY was 1.79 times higher than that in the PY (95% Cl 1.51 to 2.13, *p* < 0.001) (Table 6).

The results of ENSO showed a special trend compared to the individual results (Table 6). For every one-unit increase in the activity of La Niña during a PY, the incidence rate of enterovirus infection increased by a factor of 1.03 (95% Cl 1.03 to 1.04, *p* < 0.001), indicating a positive correlation. However, during a NPY, the factor is 0.89 (95% Cl 0.89 to 0.89, *p* < 0.001) for every one-unit increase in the activity of La Niña. In terms of the incidence rate of EVSCs, the factor is 0.77 (95% Cl 0.71 to 0.83, *p* < 0.001) during a PY and 1.94 (95% Cl 1.75 to 2.16, *p* < 0.001) during a NPY, respectively. In the aspect of the ratio of EVSCs, the factor is 0.74 (95% Cl 0.68 to 0.80, *p* < 0.001) during a PY and 2.37 (95% Cl 2.12 to 2.65, *p* < 0.001) during a NPY.

## 4. Discussion

Although previous articles have discussed the association between enterovirus and climate factors El Niño, as well as the relationship with the ENSO, there have been no articles exploring the connection between enterovirus infections and the PYF based on the “Yunqi theory”. We are the first to investigate the association between enterovirus infection and two kinds of CFCs. The concept of Yunqi theory originates from Huangdi’s Internal Classic (Huang Di Nei Jing), which is the earliest existing traditional Chinese medical work. It systematically discusses the relationship among climate, astronomy, calendars, the human body, disease, and the environment and was further used as a method for predicting epidemics for over two thousand years. Based on Yin-Yang and the Five Elements combined with the Celestial Stems (Heavenly Stems) and Terrestrial Branches (Earthly Branches), Yunqi theory has a 60-year cycle, also known as the sexagenary cycle (Table 3). Yunqi theory has been previously applied in the research of certain epidemics such as COVID-19, Rubella virus, and influenza, which reveal correlations with climate factors [20,38,39].

Our results showed that the incidence rate of enterovirus infections, incidence rate of EVSCs, and ratio of EVSCs in NPYs were significantly higher than those in PYs (*p* < 0.001). The difference in the incidence rate of EVSCs between PYs and NPYs was more significant. Although the incidence rates of enterovirus infections were lower in 2008 and 2011, both the incidence rates and the proportion of EVSCs were higher than the average. This indicates that the PYF factor not only affected the incidence rates of enterovirus infections but it was also particularly significant in the occurrence of EVSC cases. Since the major outbreak in 1998, enterovirus infections with severe complications (EVSC) have occurred approximately every 2–3 years, with a severe case fatality rate ranging from 3.8% to 25.7%. Infants and young children are at high risk of EVSC and even death. Children under 5 years of age account for approximately 91.2% of all cases of EVSC, and the younger the individual, the more severe the illness. As age increases, asymptomatic infections become more common, and the relative risk decreases [6]. In terms of medical expenses, the average daily hospitalization cost for cases of EVSC is 5.1 times higher than that before EVSC onset, compared to a 1.34-fold increase for cases with mild symptoms. Due to the relatively high mortality rate and high medical costs needed, EVSC is a noteworthy issue in overall social costs [6,40]. Previous studies have indicated that public health policies including handwashing reduced the incidence of EVSC since 2012, which is consistent with our findings (Figure 1) [41]. However, our study observed a period of several years before and after the implementation of public health policies, and we still observed a significant impact of PYF.

The higher correlation between the PYF and incidence rates of EVSCs compared to that of regular enterovirus infections can potentially be explained by the fact that the diseases described in ancient texts were severe enough to be documented and transmitted through generations [42,43,44]. Given the higher morbidity and mortality rates associated with cases of EVSC, their considerable impact on medical and societal costs became more notable [6,40]. The PYF, therefore, serves as a more prominent predictive indicator of EVSC compared to normal conditions. This could be attributed to the influence of the PYF on weather variations, as previous research has indicated that the mean temperature and diurnal temperature ranges may affect the lifetime of the virus [26,45].

Since the outbreak of COVID-19 at the end of 2019, the government has actively promoted epidemic prevention policies in addition to the development of vaccines and medications, indirectly preventing the spread of enterovirus [46,47,48]. Our results showed a significant decrease in the incidence rates of both enterovirus infections and EVSCs between 2020 and 2022, with a record of zero cases of EVSC in 2021. However, even without excluding the data from these three years, a significant difference (*p* < 0.001) was observed in the incidence rates of each group between PY and NPY. This finding suggests that, while public health policies can influence the overall incidence rate, the PYF continues to exert a substantial influence on these rates. Moreover, in terms of the EVSC ratio, NPYs (2008, 2011, and 2012) still had higher ratios than the corrected average (excluding data from 2020 to 2022), further indicating the influence of epidemic years on cases of EVSC.

Regarding ENSO, the incidence rate of enterovirus infections decreased to 0.96 times for every one-unit increase in the La Niña activity (*p*-value < 0.001) (Table 6). This means that the higher the La Niña level in ENSO was, the lower the incidence rate of enterovirus infections. That is, La Niña could be considered a protective factor. However, in both the incidence rate of EVSCs and the ratio of EVSCs, the opposite trend appeared. Although the *p*-value for the incidence rate of EVSCs was nonsignificant, the ratio of EVSCs was still significant. For every one-unit increase in the activity of La Niña, the ratio of EVSCs increased by a factor of 1.06 (*p* = 0.008). If we consider the temperature fluctuations caused by ENSO, a possible correlation between ENSO and the incidence of EVSC was observed. In years with a high incidence rate of EVSCs (2008, 2012, and 2019), the preceding one or two years were La Niña years. Furthermore, following three consecutive La Niña years, there was a relative increase in severe illness rates in 2022. In the case of Taiwan, although it is not located in the region that is primarily influenced by ENSO events, the occurrence or development of El Niño can lead to warmer temperatures in winter, resulting in a higher likelihood of a mild winter. The following spring may experience increased rainfall, and there can be warmer temperatures in the subsequent summer [49]. During the occurrence or development of La Niña events, Taiwan tends to experience increased rainfall in the eastern part of the country during autumn, with less significant effects on winter temperatures. However, during the occurrence or development of La Niña events, there is a tendency for the average temperature in the following year to decrease [49,50], which corresponds with previous statistical data according to the Central Weather Bureau of Taiwan (https://opendata.cwb.gov.tw/dataset/climate/C-B0026-001, accessed on 23 May 2023). Taiwan experienced lower average temperatures in 2008, 2011, and 2012 after the La Niña events in 2007 and 2010. Interestingly, in 2008 and 2012, there were also high incidence rates of EVSC, suggesting a potential delayed impact of La Niña events on the incidence of cases of EVSC.

By taking the PYF and ENSO into account simultaneously, we can obtain a standardized presentation without the influence of the other. The effect of the PYF showed a similar tendency in every aspect, while ENSO exhibited an opposite effect in different situations. On the one hand, if, during a PY, the incidence rate of enterovirus infections increased by one unit due to the activity of La Niña, the rate changed to 1.03, which was 0.96 in individual analysis (Table 6). In a NPY, the incidence rate of enterovirus infections is 0.89, representing a resemble effect with individual analysis. This means that the PYF was sufficient to change the trend of ENSO. The scale and complexity of the PYF as a CFC are greater than those of ENSO. On the other hand, if ENSO activity was the same, the incidence rate of enterovirus infections in NPY changed to 1.29 times compared to that in PY, which was 1.24 in individual analysis. This indicates that after considering the influence of ENSO, the trend of the PYF remains unaffected. This also suggests that although they are both CFCs, the impact of the PYF is greater.

There are several limitations in this study. First, the annual number of enterovirus emergency department visits is determined based on clinical diagnosis, which may not always involve laboratory testing. Therefore, there is a possibility of inclusion of other non-enteroviral diseases with similar clinical manifestations. Second, only the number of patients who visited the emergency department was used in this study as a representative sample. Since the population stratification for outpatient statistics in the Centers for Disease Control and Prevention database is for those under 14 years of age, it was not suitable to add outpatient data (under 15 years of age) to the emergency department data. However, the trend from 2009 to 2022 was similar to that in the emergency department; only the number of emergency department visits was used, and the year had the potential to affect the incidence rate. In the future, when collecting more annual data, it is recommended to consider including outpatient data. This study collected the incidence rates of enterovirus infections in the last 16 years, with an equal number of cases in the PY group (*n* = 8) and the NPY group (*n* = 8), and more annual data should be collected in the future.

## 5. Conclusions

This is the first study to analyze the relationship between the incidence of enterovirus infections and CFCs in Taiwan. With the PYF, the incidence rates of enterovirus infections and EVSCs and the ratio of EVSCs in non-ping-years were significantly higher than those in ping-years. ENSO was another CFC that had a relatively small but significant impact on the incidence of enterovirus infections. When considering the impact of the PYF and ENSO simultaneously, the influence of non-ping-years surpassed that of ENSO, suggesting that the PYF in the Yunqi theory is a potential predictive factor for future enterovirus outbreaks in Taiwan.

Concept Highlights:✓Enterovirus infection with severe complications (EVSC): Instead of the typical presentation of enterovirus infection like hand, foot, and mouth disease or herpangina, cases of EVSC show severe signs and symptoms. All cases of EVSC meet both clinical and laboratory criteria.✓Climate factor complex (CFC): A combination of several climate factors including temperature, humidity, rainfall, wind speed, sunshine, air pressure, or more. CFC can affect the environment in a large-scale way.✓Yunqi theory: A theory in Huangdi’s Internal Classic that is based on ten Celestial Stems (Heavenly Stems) and twelve Terrestrial Branches (Earthly Branches). Celestial Stems govern the Major Yun, while the Terrestrial Branches govern the type of Major Qi each year (Table 1 and Table 2).✓Ping-year factor (PYF): A certain combination of Celestial Stems and Terrestrial Branches used to determine whether a year is a ping-year (PY) or a non-ping-year (NPY). There is a total of 28 PYs and 32 NPYs in a cycle of 60 years (Table 3).✓El Niño-Southern Oscillation (ENSO): A phenomenon presents alternating El Niño and La Niña conditions. During the El Niño phase, the sea surface temperature and rainfall increase in the eastern Pacific region, while in the western Pacific region, the sea surface temperature and rainfall decrease. The La Niña phase exhibits a trend opposite to that of the El Niño phase.

## Figures and Tables

**Figure 1 viruses-16-00471-f001:**
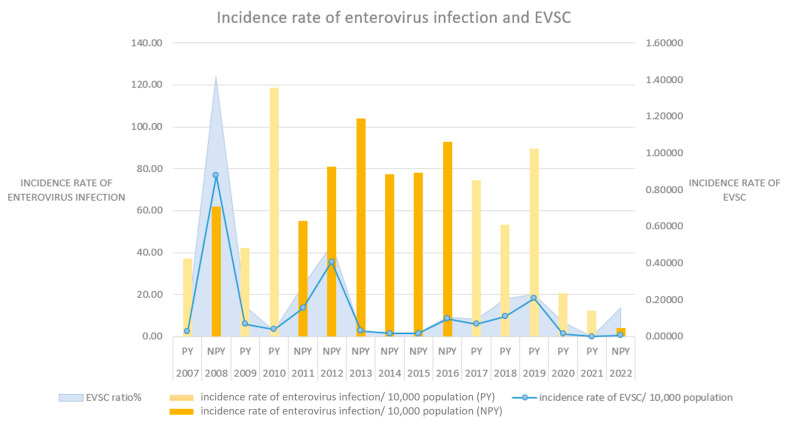
The incidence rates of enterovirus infections and EVSCs from 2007 to 2022. (PY: ping-year; NPY: non-ping-year).

**Table 1 viruses-16-00471-t001:** Major Yun in different Celestial Stem years.

Celestial Stems	Jiǎ and Jǐ (甲, 己)	Yǐ and Gēng (乙, 庚)	Bǐng and Xīn (丙, 辛)	Dīng and Rén (丁, 壬)	Wù and Guǐ (戊, 癸)
Major Yun	Earth	Metal	Water	Wood	Fire

**Table 2 viruses-16-00471-t002:** Major Qi in different Terrestrial Branch periods.

Terrestrial Branches	First Qi	Second Qi	Third Qi	Forth Qi	Fifth Qi	Sixth Qi
Zǐ and Wǔ (子, 午)	Greater Yang	Reverting Yin	Lesser Yin	Greater Yin	Lesser Yang	Yang Brightness
Chǒu and Wèi (丑, 未)	Reverting Yin	Lesser Yin	Greater Yin	Lesser Yang	Yang Brightness	Greater Yang
Yín and Shēn (寅, 申)	Lesser Yin	Greater Yin	Lesser Yang	Yang Brightness	Greater Yang	Reverting Yin
Mǎo and Yǒu (卯, 酉)	Greater Yin	Lesser Yang	Yang Brightness	Greater Yang	Reverting Yin	Lesser Yin
Chén and Xū (辰, 戌)	Lesser Yang	Yang Brightness	Greater Yang	Reverting Yin	Lesser Yin	Greater Yin
Sì and Hài (巳, 亥)	Yang Brightness	Greater Yang	Reverting Yin	Lesser Yin	Greater Yin	Lesser Yang

**Table 3 viruses-16-00471-t003:** Sexagenary cycle, combinations of Celestial stems and Terrestrial branches (from 1984 to 2043 for example). Each term consists of two Chinese characters, the first being one of the Celestial Stems and the second being one of the Terrestrial Branches. Dark plates are NPYs, and light plates are PYs.

Celestial Stems (Heavenly Stems)—Terrestrial Branches (Earthly Branches)									
Jiǎ-Zǐ 甲子 1984	Yǐ-Chǒu 乙丑 1985	Bǐng-Yín 丙寅 1986	Dīng-Mǎo 丁卯 1987	Wù-Chén 戊辰 1988	Jǐ-Sì 己巳 1989	Gēng-Wǔ 庚午 1990	Xīn-Wèi 辛未 1991	Rén-Shēn 壬申 1992	Guǐ-Yǒu 癸酉 1993
Jiǎ-Xū 甲戌 1994	Yǐ-Hài 乙亥 1995	Bǐng-Zǐ 丙子 1996	Dīng-Chǒu 丁丑 1997	Wù-Yín 戊寅 1998	Jǐ-Mǎo 己卯 1999	Gēng-Chén 庚辰 2000	Xīn-Sì 辛巳 2001	Rén-Wǔ 壬午 2002	Guǐ-Wèi 癸未 2003
Jiǎ-Shēn 甲申 2004	Yǐ-Yǒu 乙酉 2005	Bǐng-Xū 丙戌 2006	Dīng-Hài 丁亥 2007	Wù-Zǐ 戊子 2008	Jǐ-Chǒu 己丑 2009	Gēng-Yín 庚寅 2010	Xīn-Mǎo 辛卯 2011	Rén-Chén 壬辰 2012	Guǐ-Sì 癸巳 2013
Jiǎ-Wǔ 甲午 2014	Yǐ-Wèi 乙未 2015	Bǐng-Shēn 丙申 2016	Dīng-Yǒu 丁酉 2017	Wù-Xū 戊戌 2018	Jǐ-Hài 己亥 2019	Gēng-Zǐ 庚子 2020	Xīn-Chǒu 辛丑 2021	Rén-Yín 壬寅 2022	Guǐ-Mǎo 癸卯 2023
Jiǎ-Chén 甲辰 2024	Yǐ-Sì 乙巳 2025	Bǐng-Wǔ 丙午 2026	Dīng-Wèi 丁未 2027	Wù-Shēn 戊申 2028	Jǐ-Yǒu 己酉 2029	Gēng-Xū 庚戌 2030	Xīn-Hài 辛亥 2031	Rén-Zǐ 壬子 2032	Guǐ-Chǒu 癸丑 2033
Jiǎ-Yín 甲寅 2034	Yǐ-Mǎo 乙卯 2035	Bǐng-Chén 丙辰 2036	Dīng-Sì 丁巳 2037	Wù-Wǔ 戊午 2038	Jǐ-Wèi 己未 2039	Gēng-Shēn 庚申 2040	Xīn-Yǒu 辛酉 2041	Rén-Xū 壬戌 2042	Guǐ-Hài 癸亥 2043

**Table 4 viruses-16-00471-t004:** ENSO activity.

Year	ONI	ENSO	Level
2007	−1.5~1.9	La Niña	Strong
2008	−0.5~−0.9	La Niña	Weak
2009	1~1.4	El Niño	Moderate
2010	−1.5~1.9	La Niña	Strong
2011	−1~−1.4	La Niña	Moderate
2012		Neutral	
2013		Neutral	
2014	0.5~0.9	El Niño	Weak
2015	>2	El Niño	Very Strong
2016	−0.5~−0.9	La Niña	Weak
2017	−0.5~−0.9	La Niña	Weak
2018	0.5~0.9	El Niño	Weak
2019		Neutral	
2020	−1~−1.4	La Niña	Moderate
2021	−1~−1.4	La Niña	Moderate
2022	−0.5~−0.9	La Niña	Weak

**Table 5 viruses-16-00471-t005:** Results of enterovirus infections/EVSCs, the total population, the incidence rate of enterovirus infections/EVSCs, and the ratio of EVSCs in each year.

Year	Ping-Year Factor	ENSO	ENSO Level	No. of Enterovirus Infections	No. of EVSCs	Total Population	Incidence Rate of Enterovirus Infections *	Incidence Rate of EVSCs *	Ratio of EVSCs
2007	PY	La Niña	Strong	16,192	12	4,350,461	37.22	0.028	0.074
2008	NPY	La Niña	Weak	26,223	371	4,231,147	61.98	0.877	1.415
2009	PY	El Niño	Moderate	17,264	28	4,100,007	42.11	0.068	0.162
2010	PY	La Niña	Strong	46,933	16	3,948,315	118.87	0.041	0.034
2011	NPY	La Niña	Moderate	21,091	59	3,823,867	55.16	0.154	0.28
2012	NPY	Neutral		30,289	152	3,734,674	81.1	0.407	0.502
2013	NPY	Neutral		37,560	11	3,613,842	103.93	0.03	0.029
2014	NPY	El Niño	Weak	27,566	6	3,559,994	77.43	0.017	0.022
2015	NPY	El Niño	Very Strong	27,311	6	3,493,764	78.17	0.017	0.022
2016	NPY	La Niña	Weak	31,548	33	3,398,892	92.82	0.097	0.105
2017	PY	La Niña	Weak	24,820	23	3,338,143	74.35	0.069	0.093
2018	PY	El Niño	Weak	17,510	36	3,275,623	53.46	0.11	0.206
2019	PY	Neutral		28,967	67	3,228,875	89.71	0.208	0.231
2020	PY	La Niña	Moderate	6546	5	3,171,193	20.64	0.016	0.076
2021	PY	La Niña	Moderate	3790	0	3,094,757	12.25	0	0
2022	NPY	La Niña	Weak	1281	2	3,022,295	4.24	0.007	0.156
Average							62.72	0.134	0.213
Corrected average							74.33	0.163	0.244

*: per 10,000 population.

**Table 6 viruses-16-00471-t006:** The impact of the PYF and ENSO on the incidence of enterovirus infections. Ping-year (PY) and non-ping-year (NPY) are defined by the PYF. La Niña condition is set as the index of ENSO. The left side shows the analysis of the PYF and ENSO separately; the right side shows the analysis of the PYF and ENSO combined.

Separate PYF and ENSO					Combined PYF and ENSO				
Predictors	Incidence Rate Ratio	95% CI		*p* < Value	Predictors	Incidence Rate Ratio	95% CI		*p*-Value
		Lower	Upper				Lower	Upper	
incidence rate of enterovirus infections									
PY	1	-	-	-	PY	1	-	-	-
NPY	1.24	1.23	1.24	<0.001 ***	NPY	1.29	1.28	1.29	<0.001 ***
La Niña	0.96	0.96	0.97	<0.001 ***	La Niña (PY)	1.03	1.03	1.04	<0.001 ***
					La Niña (NPY)	0.89	0.89	0.89	<0.001 ***
incidence rate of EVSCs									
PY	1	-	-	-	PY	1	-	-	-
NPY	3.38	2.88	3.99	<0.001 ***	NPY	2.38	2.00	2.83	<0.001 ***
La Niña	1.02	0.98	1.06	0.393	La Niña (PY)	0.77	0.71	0.83	<0.001 ***
					La Niña (NPY)	1.94	1.75	2.16	<0.001 ***
ratio of EVSCs									
PY	1	-	-	-	PY	1	-	-	-
NPY	2.73	2.33	3.23	<0.001 ***	NPY	1.79	1.51	2.13	<0.001 ***
La Niña	1.06	1.01	1.10	0.008 **	La Niña (PY)	0.74	0.68	0.80	<0.001 ***
					La Niña (NPY)	2.37	2.12	2.65	<0.001 ***

**: Correlation is significant at the 0.01 level. ***: Correlation is significant at the 0.001 level.

## Data Availability

All data generated or analyzed during this study are included in this published article and its Appendix A.

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
