# Peer review of "Investigation of the Correlation between Enterovirus Infection and the Climate Factor Complex Including the Ping-Year Factor and El Niño-Southern Oscillation in Taiwan"

_viruses, 2024, doi:10.3390/v16030471_

Round 1

Reviewer 1 Report

Comments and Suggestions for Authors

Reviewer Comments

Manuscript number: Viruses-2883131

Title: Investigation of the correlation between enterovirus infection and the climate factor complex including the Ping-Year factor and El Nino-Southern Oscillation in Taiwan.

Summary

Enteroviruses are endemic globally and a major cause of morbidity and mortality. In this manuscript the authors explore the capacity of climate factor complexes (CFC) like ping-year factor (PYF) and the El Nino-southern Oscillation (ENSO) (alone or in combination) to predict incidence of enterovirus infections and/or severe complications. They found that these CFCs could be potential predictors of enterovirus infection.

Recommendation

The authors have done a good job, and their study deserves to be published as it adds to the body of knowledge on strategies for anticipating and predicting likely peaks in enterovirus infections and associated severe complications. However, enteroviruses are a broad group, and the authors might need to better define the enterovirus types referenced in the database explored as data source for this study. The study deserves to be published with minor revision.

General Comments

1.     Does enterovirus as defined here include rhinoviruses? Please consider being more precise in your definition. What does the data really capture? HFMD?

2.     In section 2.1 (Data collection) please (if possible) clearly state how enterovirus infection was diagnosed. Specifically, what assays (5’UTR, VP1, complete genome, Cell culture, neutralization assay) were used to confirm enteroviruses were present in the samples.

3.     In section 2.1.1 to 2.1.3 (EVSC case definition) please (if possible) clearly state how enterovirus infection was diagnosed. Specifically, what assays (5’UTR, VP1, complete genome, Cell culture, neutralization assay) were used to confirm enteroviruses were present in the samples.

4.     Page 8, Lines 281-282: Please revisit the sentence, it seems incomplete. Either a word is missing, or a word is present that should have been deleted.

5.     Page 8, Lines 310-311: Please revisit the sentence, it seems incomplete. Either a word is missing, or a word is present that should have been deleted.

Author Response

  1. Does enterovirus as defined here include rhinoviruses? Please consider being more precise in your definition. What does the data really capture? HFMD?

Response:

Thank you for kindly suggesting that we clarify the definition. The database used in our study was collected from the Taiwan National Infectious Disease Statistics System on the website of the Taiwan Centers for Disease Control (Tw-CDC). The data of annual enterovirus cases were transferred from the National Health Insurance research database of Taiwan by using ICD-9-CM/ICD-10 codes. Herpangina and HFMD (ICD-9-CM of 074.3 or 074.0, and ICD-10 of B08.4 or B08.5 respectively) were diagnosed mainly based on clinical symptoms and signs, with or without laboratory examination. [1] All confirmed cases of EVSC meet both clinical and laboratory criteria.

  1. Clinical criteria: any one of the following three criteria
  • Typical symptoms of hand, foot, and mouth disease or herpangina, with the presence of myoclonic jerks, or complications such as encephalitis, acute flaccid paralysis syndrome, acute hepatitis, myocarditis, acute cardiomyopathy, heart and lung failure, and other severe conditions.
  • Absence of hand, foot, and mouth disease or herpangina, but with respiratory infection symptoms and suspected enterovirus infection accompanied by brainstem encephalitis or acute flaccid myelitis.
  • Infants under three months of age who presented with symptoms of sepsis, such as myocarditis, hepatitis, encephalitis, thrombocytopenia, and multiple organ failure, and excluding other common bacterial infections.
  1. Laboratory criteria: any one of the following three criteria
  • Clinical specimens (throat swabs or throat washes, feces or rectal swabs, cerebrospinal fluid or vesicle fluid, etc.) were isolated and identified as enterovirus.
  • Molecular biological nucleic acid testing of clinical specimens was positive.
  • Serological antibody testing was positive (referring to specific IgM antibodies against enterovirus type 71 in serum)

The virologic surveillance data reported from Tw-CDC includes coxsackievirus groups A/B (CA/CB), echovirus, EV71, EV-D68, and other enteroviruses. “Other enteroviruses” refers to all serotypes that were not among the top 5 mentioned and were not individually listed. [2] Rhinovirus and human parechovirus may be included in “other enteroviruses” among annual enterovirus cases, but their numbers are few. They were excluded from the cases of EVSC. Above definition of the enterovirus used here have been added in line 43-47, 53-57, 126-130, and 150-168.

  1. In section 2.1 (Data collection) please (if possible) clearly state how enterovirus infection was Specifically, what assays (5’UTR, VP1, complete genome, Cell culture, neutralization assay) were used to confirm enteroviruses were present in the samples.

Response:

The data of annual enterovirus cases were transferred from the National Health Insurance research database of Taiwan by using ICD-9-CM/ICD-10 codes. Herpangina and HFMD (ICD-9-CM of 074.3 or 074.0, and ICD-10 of B08.4 or B08.5 respectively) were diagnosed mainly based on clinical symptoms and signs, with or without laboratory examination. [1] The above sentences have been added in line 126-129.

  1. In section 2.1.1 to 2.1.3 (EVSC case definition) please (if possible) clearly state how enterovirus infection was diagnosed. Specifically, what assays (5’UTR, VP1, complete genome, Cell culture, neutralization assay) were used to confirm enteroviruses were present in the samples.

Response:

Thank you for suggestion. All confirmed cases of enterovirus infection with severe complications (EVSC) meet both clinical and laboratory criteria.

  1. Clinical criteria: any one of the following three criteria
  • Typical symptoms of hand, foot, and mouth disease or herpangina, with the presence of myoclonic jerks, or complications such as encephalitis, acute flaccid paralysis syndrome, acute hepatitis, myocarditis, acute cardiomyopathy, heart and lung failure, and other severe conditions.
  • Absence of hand, foot, and mouth disease or herpangina, but with respiratory infection symptoms and suspected enterovirus infection accompanied by brainstem encephalitis or acute flaccid myelitis.
  • Infants under three months of age who presented with symptoms of sepsis, such as myocarditis, hepatitis, encephalitis, thrombocytopenia, and multiple organ failure, and excluding other common bacterial infections.
  1. Laboratory criteria: any one of the following three criteria
  • Clinical specimens (throat swabs or throat washes, feces or rectal swabs, cerebrospinal fluid or vesicle fluid, etc.) were isolated and identified as enterovirus.
  • Molecular biological nucleic acid testing of clinical specimens was positive.
  • Serological antibody testing was positive (referring to specific IgM antibodies against enterovirus type 71 in serum)

The methods used by the contracted laboratories of the Tw-CDC for viral infections include viral culture, immunofluorescence assay, and polymerase chain reaction (EV CODEHOP RT–snPCR or pan-enterovirus real-time RT-PCR, target gene: 5'UTR and VP1). If the type of enterovirus could not be differentiated, further Sanger sequencing would be arranged. [3] We have added the diagnostic criteria including specific laboratory methods to the manuscript in section 2.1 line 150-168.

  1. Page 8, Lines 281-282: Please revisit the sentence, it seems incomplete. Either a word is missing, or a word is present that should have been deleted.

Response:

The sentence has been revised in line 337-340.

  1. Page 8, Lines 310-311: Please revisit the sentence, it seems incomplete. Either a word is missing, or a word is present that should have been deleted.

Response:

The sentence has been revised in line 369-372.

References:

  1. Kuo SC, Tsou HH, Wu HY, Hsu YT, Lee FJ, Shih SM, Hsiung CA, Chen WJ. Nonpolio Enterovirus Activity during the COVID-19 Pandemic, Taiwan, 2020. Emerg Infect Dis. 2021 Jan;27(1):306–8. doi: 10.3201/eid2701.203394. Epub 2020 Dec 1. PMID: 33261719; PMCID: PMC7774568.
  2. Tseng FC, Huang HC, Chi CY, Lin TL, Liu CC, Jian JW, Hsu LC, Wu HS, Yang JY, Chang YW, Wang HC, Hsu YW, Su IJ, Wang JR; CDC-Taiwan Virology Reference Laboratories and Sentinel Physician Network. Epidemiological survey of enterovirus infections occurring in Taiwan between 2000 and 2005: analysis of sentinel physician surveillance data. J Med Virol. 2007 Dec;79(12):1850-60. doi: 10.1002/jmv.21006. PMID: 17935170.
  3. Yang JY, L.C., Development of reagent kits for enterovirus molecular diagnosis. Department of Disease Control and Prevention, Ministry of Health and Welfare, 2020. https://www.grb.gov.tw/search/planDetail?id=12878301

Reviewer 2 Report

Comments and Suggestions for Authors

This study showed that the relations of a climate factor complex (CFC), the ping-year factor (PYF) and the incidence of enterovirus infection of children aged 15 years and younger with enterovirus infection enrolled between 2007 and 2022.

It is very interesting and useful for the prediction of enterovirus infection in Taiwan.

But, the explanation of ping-year factor (PYF) is not enough in introduction and discussion. I cannot understand this factor clearly.

Please explain clearly ping-year factor (PYF).

If the explanation of this factor is not clear, please delete this ping-year factor (PYF) in this study.

Author Response

Reviewer #2:

Please explain clearly ping-year factor (PYF).

Response: 

Thank you for kindly suggesting that we clarify the explanation.

Ping-year factor (PYF) is an indicator originated from Yunqi theory. Yunqi theory is one of a theory in Huangdi’s Internal Classic that based on ten Celestial Stems (Heavenly Stems) and twelve Terrestrial Branches (Earthly Branches). The Celestial Stems govern the Major Yun, while the Terrestrial Branches govern the type of Major Qi of each year. [1,2,3] Depending on the different characteristics from the interaction of the Celestial Stems and Terrestrial Branches of each year, the climate changes and the manifestations of corresponding diseases are determined. Thus, PYF is a certain combination of Celestial Stems and Terrestrial Branches used to determine whether a year is a ping-year or non-ping-year. There is a total of 28 ping-years and 32 non-ping-years in a cycle of 60 years. Above explanation of PYF have been added in the introduction in line 79-89, 90-92, and section 2.3.1 in line 186-190.  

References:

  1. Zhang DS, He J, Gao SH, Hu BK, Ma SL. Correlation analysis for the attack of respiratory diseases and meteorological factors. Chin J Integr Med. 2011 Aug;17(8):600-6. doi: 10.1007/s11655-011-0821-0. Epub 2011 Aug 9. PMID: 21826594.
  2. Zhang DS, Zhang X, Ouyang YH, Zhang L, Ma SL, He J. Incidence of allergic rhinitis and meteorological variables: Non-linear correlation and non-linear regression analysis based on Yunqi theory of chinese medicine. Chin J Integr Med. 2016 Jun 21. doi: 10.1007/s11655-016-2588-9. Epub ahead of print. PMID: 27329149.
  3. Zhang X, Ma SL, Liu ZD, He J. Correlation Analysis of Rubella Incidence and Meteorological Variables Based on Chinese Medicine Theory of Yunqi. Chin J Integr Med. 2019 Dec;25(12):911-916. doi: 10.1007/s11655-018-3016-0. Epub 2018 Nov 22. PMID: 30467697; PMCID: PMC7089232.

Reviewer 3 Report

Comments and Suggestions for Authors

Please see attached pdf for my comments.

Author Response

Reviewer #3:

  1. The section on PYF and enterovirus infection provides a clear overview of the study's findings on the relationship between PYF and the incidence of enterovirus infections. Figure 1 and Table 2 describe the annual number of cases, total population, incidence 195 rates of both enterovirus infections and EVSCs, and the ratio of EVSCs over the years. However, providing more explanation for terms such as EVSCs, ping-year, and non-ping-year may be helpful.

Response:

Thank you for suggestion.

EVSC:

All confirmed cases of enterovirus infection with severe complications (EVSC) meet both clinical and laboratory criteria.

  1. Clinical criteria: any one of the following three criteria
  • Typical symptoms of hand, foot, and mouth disease or herpangina, with the presence of myoclonic jerks, or complications such as encephalitis, acute flaccid paralysis syndrome, acute hepatitis, myocarditis, acute cardiomyopathy, heart and lung failure, and other severe conditions.
  • Absence of hand, foot, and mouth disease or herpangina, but with respiratory infection symptoms and suspected enterovirus infection accompanied by brainstem encephalitis or acute flaccid myelitis.
  • Infants under three months of age who presented with symptoms of sepsis, such as myocarditis, hepatitis, encephalitis, thrombocytopenia, and multiple organ failure, and excluding other common bacterial infections.
  1. Laboratory criteria: any one of the following three criteria
  • Clinical specimens (throat swabs or throat washes, feces or rectal swabs, cerebrospinal fluid or vesicle fluid, etc.) were isolated and identified as enterovirus.
  • Molecular biological nucleic acid testing of clinical specimens was positive.
  • Serological antibody testing was positive (referring to specific IgM antibodies against enterovirus type 71 in serum)

The virologic surveillance data reported from Tw-CDC includes coxsackievirus groups A/B (CA/CB), echovirus, EV71, EV-D68, and other enteroviruses. “Other enteroviruses” refers to all serotypes that were not among the top 5 mentioned and were not individually listed. [1] Rhinovirus and human parechovirus may be included in “other enteroviruses” among annual enterovirus cases, but their numbers are few. They were excluded from the cases of EVSC. Above definition of the enterovirus used here have been added in line 43-47, 53-57, 126-130, and 150-168.

ping-year, and non-ping-year:

Each year can be categorized as a ping-year (PY) or non-ping-year (NPY) according to the ping-year factor (PYF). PYF is an indicator originated from Yunqi theory. Yunqi theory is one of a theory in Huangdi’s Internal Classic that based on ten Celestial Stems (Heavenly Stems) and twelve Terrestrial Branches (Earthly Branches). The Celestial Stems govern the Major Yun, while the Terrestrial Branches govern the type of Major Qi of each year. [2.3.4] Depending on the different characteristics from the interaction of the Celestial Stems and Terrestrial Branches of each year, the climate changes and the manifestations of corresponding diseases are affected. Thus, PYF is a certain combination of Celestial Stems and Terrestrial Branches used to determine whether a year is a ping-year or non-ping-year. The term "ping-year" (PY) originates from the Chinese word "氣之平" (píng of Qi ), which conveys the meanings of stability and peace of Qi. There are a total of 28 ping-years and 32 non-ping-years in a cycle of 60 years. Above explanation of PYF have been added in the introduction in line 79-89, 90-92, and section 2.3.1 in line 186-190.

  1. The section on the ENSO, and enterovirus infection provides clear results indicating that for every one-unit increase in the activity of La Niña, the incidence rate of enterovirus infection decreased to 0.96 times, which means that the higher the La Niña level in ENSO, the lower the incidence rate of enterovirus infection (Table 3). There was an opposite trend in both the incidence rate of EVSCs and the ratio of EVSCs. However, some of the terminology used may be technical and difficult to understand. For example, please define La Niña so the reader does not have to go back multiple times to find its meaning.

Response:

Thank you for pointing this out.

ENSO is a quasiperiodic phenomenon characterized by alternating El Niño and La Niña conditions. Oceanic Niño Index (ONI) is used to represent the activity level of ENSO. The ONI tracks the continuous 3-month average sea surface temperatures compared to the 1981-2010 average in the east-central tropical Pacific between 120°-170°W. ONI values of +0.5 or higher (positive ONI) were considered El Niño conditions, values between +0.5 and -0.5 were considered neutral conditions, and values of -0.5 or lower (negative ONI) were considered La Niña conditions. During the El Niño phase, the sea surface temperature and rainfall increase in the eastern Pacific region, while in the western Pacific region, the sea surface temperature and rainfall decrease. The La Niña phase exhibits a trend opposite to that of the El Niño phase. The above sentences have been added in the section 2.3.2 in line 207-209. We have also included a “Concept Highlight” section in the manuscript (line 441-460) to provide clear and concise explanations for specialized terminology.

  1. Providing additional explanation and context for terms such as "Yunqi theory" and "CFCs" may benefit readers unfamiliar with these concepts.

Response:

Thank you for kindly suggesting that we clarify the explanation.

Additional explanations and context for terms are listed below:

Yunqi theory: A theory in Huangdi’s Internal Classic that based on ten Celestial Stems (Heavenly Stems) and twelve Terrestrial Branches (Earthly Branches). Celestial Stems govern the Major Yun, while the Terrestrial Branches govern the type of Major Qi of each year (Table 1 and 2).

Climate factor complex (CFC): A combination of several climate factors including temperature, humidity, rainfall, wind speed, sunshine, air pressure or more. CFC can affect environment in a large-scale way.

Enterovirus infection with severe complications (EVSC): Instead of typical presentation of enterovirus infection like hand, foot, and mouth disease or herpangina, cases of EVSC show severe signs and symptoms. All cases of EVSC meet both clinical and laboratory criteria.

Ping-year factor (PYF): A certain combination of Celestial Stems and Terrestrial Branches used to determine whether a year is a ping-year (PY) or a non-ping-year (NPY). There is a total of 28 PYs and 32 NPYs in a cycle of 60 years.

El Niño-Southern Oscillation (ENSO): A phenomenon presents alternating El Niño and La Niña conditions. During the El Niño phase, the sea surface temperature and rainfall increase in the eastern Pacific region, while in the western Pacific region, the sea surface temperature and rainfall decrease. The La Niña phase exhibits a trend opposite to that of the El Niño phase.

We have also included a “Concept Highlight” section in the manuscript (line 441-460) to provide clear and concise explanations for specialized terminology.

Reference

  1. Tseng FC, Huang HC, Chi CY, Lin TL, Liu CC, Jian JW, Hsu LC, Wu HS, Yang JY, Chang YW, Wang HC, Hsu YW, Su IJ, Wang JR; CDC-Taiwan Virology Reference Laboratories and Sentinel Physician Network. Epidemiological survey of enterovirus infections occurring in Taiwan between 2000 and 2005: analysis of sentinel physician surveillance data. J Med Virol. 2007 Dec;79(12):1850-60. doi: 10.1002/jmv.21006. PMID: 17935170.
  2. Zhang DS, He J, Gao SH, Hu BK, Ma SL. Correlation analysis for the attack of respiratory diseases and meteorological factors. Chin J Integr Med. 2011 Aug;17(8):600-6. doi: 10.1007/s11655-011-0821-0. Epub 2011 Aug 9. PMID: 21826594.
  3. Zhang DS, Zhang X, Ouyang YH, Zhang L, Ma SL, He J. Incidence of allergic rhinitis and meteorological variables: Non-linear correlation and non-linear regression analysis based on Yunqi theory of chinese medicine. Chin J Integr Med. 2016 Jun 21. doi: 10.1007/s11655-016-2588-9. Epub ahead of print. PMID: 27329149.
  4. Zhang X, Ma SL, Liu ZD, He J. Correlation Analysis of Rubella Incidence and Meteorological Variables Based on Chinese Medicine Theory of Yunqi. Chin J Integr Med. 2019 Dec;25(12):911-916. doi: 10.1007/s11655-018-3016-0. Epub 2018 Nov 22. PMID: 30467697; PMCID: PMC7089232.

Round 2

Reviewer 2 Report

Comments and Suggestions for Authors

This study is “Investigation of the correlation between enterovirus infection and the climate factor complex including the Ping-Year factor and El Nino-Southern Oscillation in Taiwan.”

You have immediate revised seriously and sincerely well. Therefore, it is the time to accept for publication in Viruses.